The rise of feathered dinosaurs: Kulindadromeus zabaikalicus, the oldest dinosaur with ‘feather-like’ structures

Cincotta Aude aude.cincotta@ucc.ie 1 2 3
Pestchevitskaya Ekaterina B. 4
Sinitsa Sofia M. 5
Markevich Valentina S. 6
Debaille Vinciane 7
Reshetova Svetlana A. 5
Mashchuk Irina M. 8
Frolov Andrei O. 8
Gerdes Axel 9
Yans Johan 2
Godefroit Pascal 1
1 Directorate ‘Earth and History of Life’, Royal Belgian Institute of Natural Sciences , Brussels , Belgium
2 Department of Geology, Institute of Life, Earth and Environment, University of Namur , Namur , Belgium
3 School of Biological, Earth and Environmental Sciences, University College Cork , Cork , Ireland
4 Institute of Petroleum Geology and Geophysics. AA Trofimuk , Novosibirsk , Russia
5 Institute of Natural Resources, Ecology, and Cryology, Siberian Branch of the Russian Academy of Sciences , Chita , Russia
6 Federal Scientific Center of the East Asia Terrestrial Biodiversity, Far East Branch of the Russian Academy of Sciences , Vladivostok , Russia
7 Laboratoire G-Time, Université Libre de Bruxelles , Brussels , Belgium
8 Institute of Earth’s Crust, Siberian Branch of the Russian Academy of Sciences , Irkutsk , Russia
9 Institut für Geowissenschaften, Johann Wolfgang Goethe Universität Frankfurt am Main , Frankfurt , Germany
Farke Andrew
Electronic publication date: 2019 Feb 1
Publication date: 2019
Volume: 7
Electronic Location ID: e6239
Received 2018 Oct 12; Accepted 2018 Dec 8
Copyright: ©2019 Cincotta et al.
Copyright year: 2019
Copyright holder: Cincotta et al.
License: This is an open access article distributed under the terms of the Creative Commons Attribution License, which permits unrestricted use, distribution, reproduction and adaptation in any medium and for any purpose provided that it is properly attributed. For attribution, the original author(s), title, publication source (PeerJ) and either DOI or URL of the article must be cited.
License URL: https://creativecommons.org/licenses/by/4.0/

Keywords: Feathered dinosaurs, Palynology, U-Pb dating, Jurassic of Siberia

Funding: Fonds National de la Recherche Scientifique (FRIA) Russian Foundation for Basic Research 17-04-01582 Russian Science Foundation 18-17-00038 Siberian Branch of the Russian Academy of Science IX.137.1 European Research Council StG “ISoSyC” 336718 This work was supported by the Fonds National de la Recherche Scientifique (FRIA grant), the Russian Foundation for Basic Research (No. 17-04-01582), the Russian Science Foundation (No. 18-17-00038), the Siberian Branch of the Russian Academy of Science (program IX.137.1) and the European Research Council (StG “ISoSyC”, 336718). The funders had no role in study design, data collection and analysis, decision to publish, or preparation of the manuscript.

==============================
Diverse epidermal appendages including grouped filaments closely resembling primitive feathers in non-avian theropods, are associated with skeletal elements in the primitive ornithischian dinosaur Kulindadromeus zabaikalicus from the Kulinda locality in south-eastern Siberia. This discovery suggests that “feather-like” structures did not evolve exclusively in theropod dinosaurs, but were instead potentially widespread in the whole dinosaur clade. The dating of the Kulinda locality is therefore particularly important for reconstructing the evolution of “feather-like” structures in dinosaurs within a chronostratigraphic framework. Here we present the first dating of the Kulinda locality, combining U-Pb analyses (LA-ICP-MS) on detrital zircons and monazites from sedimentary rocks of volcaniclastic origin and palynological observations. Concordia ages constrain the maximum age of the volcaniclastic deposits at 172.8 ± 1.6 Ma, corresponding to the Aalenian (Middle Jurassic). The palynological assemblage includes taxa that are correlated to Bathonian palynozones from western Siberia, and therefore constrains the minimum age of the deposits. The new U-Pb ages, together with the palynological data, provide evidence of a Bathonian age—between 168.3 ± 1.3 Ma and 166.1 ± 1.2 Ma—for Kulindadromeus. This is older than the previous Late Jurassic to Early Cretaceous ages tentatively based on local stratigraphic correlations. A Bathonian age is highly consistent with the phylogenetic position of Kulindadromeus at the base of the neornithischian clade and suggests that cerapodan dinosaurs originated in Asia during the Middle Jurassic, from a common ancestor that closely looked like Kulindadromeus. Our results consequently show that Kulindadromeus is the oldest known dinosaur with “feather-like” structures discovered so far.

Introduction

In 2010, a new Konservat-Lagerstätte was discovered in the Kulinda locality (south-eastern Siberia, Russia) by geologists from the Institute of Natural Resources, Ecology, and Cryology, SB RAS (Chita, Russia). The site has yielded numerous bones and associated integumentary structures belonging to the primitive ornithischian dinosaur, Kulindadromeus zabaikalicus (Godefroit et al., 2014). The soft-tissue remains include well-preserved skin, epidermal scales, and three types of integumentary filaments all tentatively interpreted as feathers (see the 3D reconstruction of the specimen in Fig. S1. Monofilaments in ornithischian dinosaurs were previously reported in the basal ceratopsian Psittacosaurus (Mayr et al., 2002) and in the heterodontosaurid Tianyulong (Zheng et al., 2009). However, the diversity and complexity of the elongated and compound integumentary structures in Kulindadromeus suggest that feather-like structures were likely widespread within the whole dinosaur clade and potentially present in their last common ancestor (Godefroit et al., 2014). According to Alifanov & Saveliev (2014) and Alifanov & Saveliev (2015), the dinosaur fauna at Kulinda comprises three new taxa: the ‘hypsilophodontian’ ornithopods Kulindapteryx ukureica and Daurosaurus olovus (Alifanov & Saveliev, 2014), and the ‘nqwebasaurid’ ornithomimosaur Lepidocheirosaurus natalis (Alifanov & Saveliev, 2015). However, we consider these three taxa as nomina dubia, and very likely synonyms of Kulindadromeus zabaikalicus (see Supplementary Information for a brief discussion on the composition of the Kulinda dinosaur fauna).

The stratigraphic section at Kulinda belongs to the base of the Ukurey Formation in the Olov Depression. K-Ar radiometric analyses on basalts and rhyolites from the base of the Ukurey Fm proposed ages between 153–157 Ma (Kimmeridgian) and 147–165 Ma (Callovian-Berriasian), respectively (Sinitsa, 2011a; Sinitsa, 2011b). Palaeo entomological and microfaunal comparisons with the Glushkovo Formation in the Unda-Daya Depression also suggested a Late Jurassic—Early Cretaceous age for the Ukurey Formation (Sinitsa & Starukhina, 1986; Sinitsa, 2011a; Sinitsa, 2011b). The age of the Kulinda deposits has not been directly investigated so far.

Here, we refine the age of the Kulinda locality using U-Pb dating of zircons and comparisons of the palynomorph and megafloral assemblages collected from the volcaniclastic layers that have also yielded the Kulindadromeus fossils. Dating of the Kulinda locality is particularly important for a better timing of the evolutionary history of integumentary structures, including feathers, in dinosaurs. Thus, the oldest well-dated integumentary structures in dinosaurs are from paravian theropods (Anchiornis, Xiaotingia, Eosinopteryx, Aurornis, and Serikornis (Hu et al., 2009; Xu et al., 2011; Godefroit et al., 2013a; Godefroit et al., 2013b; Lefèvre et al., 2017)) and in the heterodontosaurid Tianyulong (Zheng et al., 2009) from the Tiaojishan Formation, in the Daxishan section near the town of Linglongta, Jianchang County in western Liaoning Province (China). Recent U-Pb analyses have dated this section as Oxfordian (Late Jurassic), with an age ranging between 160.254 ± 0.045 Ma and 160.889 ± 0.069 Ma (Chu et al., 2016). The present paper describes the depositional history of the Kulinda section and provides data about the age of the neornithischians and the implication of this for the early evolution of feathers in dinosaurs.

Geological Setting

The Kulinda locality (Fig. 1A) is situated between two major fault zones related to the closure of the Mongol-Okhotsk Ocean that likely took place in south-eastern Siberia at the Early-Middle Jurassic boundary (Zorin, 1999; Zorin et al., 2001), or during the early Middle Jurassic (Jolivet et al., 2013; see also Supplementary Text). Both fault zones delimit a series of grabens in the area. The excavated sections at Kulinda belong to the lower part of the Ukurey Formation that crops out in the Olov Valley and several other isolated depressions in the central and south-eastern Transbaikal region (Rudenko & Starchenko, 2010). The formation is composed of interbedded sandstones, tuffaceous sandstones, conglomerates, tuffaceous conglomerates, siltstones, breccia, andesites, basaltic trachyandesites, basaltic andesites, and tuffs, up to a thickness of 850 metres (Anashkina et al., 1997). The geological map of the Transbaikal region indicates that Upper Jurassic volcaniclastic deposits crop out in Kulinda area (Fig. 1B). Field work conducted in this area and described in 2011 in a local—unpublished—report identified the remains of a volcanic edifice, named “Pharaoh”, ca. five kilometers south of the Kulinda locality (Kozlov, 2011). That structure is still clearly visible in the landscape (Fig. S2). The report notes that trachyandesites from the Pharaoh volcano have been collected on the left bank of the Olov River, and were dated at 180 ± 5 and 188 ± 6 Ma by K-Ar methods (Kozlov, 2011). In the same report, K-Ar dating of other volcanic rocks, this time collected on the right bank of the Olov River, indicate younger ages (155 ± 5.0 Ma, 104 ± 3.0 Ma, and 103 ± 4.0 Ma). The wide age range reported for the volcanic rocks collected on the Pharaoh complex suggests several volcanic episodes in the area from the Early Jurassic up to the Early Cretaceous.

Figure 1 Location of the studied area.

(A) Position of Kulinda locality with respect to the Mongol-Okhotsk suture (modified with permission from Tomurtogoo et al., 2005). (B) Geological map of the Kulinda region (redrawn with permission from Kozlov, Zaikov & Karasev, 1998). According to the map, Kulinda is situated in the Upper Jurassic of the Ukurey Formation.

Felsic igneous rocks are exposed on top of the Kulinda hill and consist of granites, biotite granites, and biotite-quartz monzonites (see Table S1 and Fig. S3). These plutonic rocks likely constitute an outcropping part of the basement. The Kulinda stratigraphical section mainly consists of a heterogeneous succession of volcaniclastic deposits (Fig. 2A). It comprises: (1) thinly laminated mudstones, (2) a single layer of tuffaceous siltstones showing glass shards, (3) lithic arenites (sandstones) including reworked fragments mainly of volcanic origin, (4) greywackes (matrix-supported) and feldspathic arenites (grain-supported) of silt- and sand-sizes, and (5) coarse- grained to brecciated sandstones mainly composed of automorph quartz and feldspars (see Fig. S4 and S5). Excavations at Kulinda consisted of three parallel trenches located at successive altitudes on the southern slope of the hill (Fig. 2B). The rock layers dip 20–30° to the south. The trenches are not correlated laterally due to their poor exposure. The vertical stratigraphic distances were therefore estimated by means of rock dip and horizontal distances between the trenches, and correspond to ca. 11–17 m between trench 4 and trench 3/3, and ca. 36–58 m between trench 3/3 and trench 3. Deposits from trench 4 are considered to be the oldest, being on the lowest part of the slope (altitude 680 m), deposits from trench 3/3 are intermediate (alt. 690 m), and those from trench 3 are located higher up on the hill (alt. 720 m; Figs. 2B–2C). Trench 4 and trench 3 are laterally separated by about 130 metres (Fig. 2B).

Figure 2 Lithological section of the Kulinda dinosaur locality in the Ukurey Formation.

(A) Composite stratigraphic log of the three trenches and the position of the bone beds. (B) Schematic location of the trenches excavated on the hillslope. (C) Photograph showing the location of the three parallel trenches on site (photo credit: A. Cincotta). The locations of the detrital samples used for dating are marked by an arrow.

Materials & Methods

U-Pb geochronology

Zircons were sampled from volcaniclastic sandstones in two different locations: (1) in the lowest part of trench 4, beneath the bone bed 4, and (2) in the medial part of trench 3, beneath the bone bed 3. Both zircons and monazites were sampled in granites cropping out on top of the Kulinda hill. The samples were prepared for mineral separation at the ‘Laboratoire G-Time’ (Université Libre de Bruxelles, Brussels). The rocks were previously fragmented by Selfrag high voltage pulse to liberate intact grains. Zircons and monazites were then separated by standard methods using heavy liquids, hand-picked under a binocular microscope, mounted on epoxy resin, and eventually polished.

Uranium, thorium and lead isotope analyses were carried out by laser ablation-inductively coupled plasma-mass spectrometry (LA-ICP-MS) at the Goethe University of Frankfurt (GUF), using a slightly modified method, described in (Gerdes & Zeh, 2006; Gerdes & Zeh, 2009). A Thermo Scientific Element 2 sector field ICP-MS was coupled to a Resolution S-155 (Resonetics) 193 nm ArF Excimer laser (CompexPro 102, Coherent) equipped with two-volume ablation cell (Laurin Technic, Australia). The laser was fired with 5.5 Hz at a fluence of about 2–3 J cm−2. The above configuration, with a spot size of 30 µm and depth penetration of 0.6 µm s−1, yielded a sensitivity of 11,000–14,000 cps/ppm 238U. Raw data were corrected offline for background signal, common Pb, laser induced elemental fractionation, instrumental mass discrimination, and time-dependent elemental fractionation of Pb/U using an in-house MS Excel  spread sheet program (Gerdes & Zeh, 2006; Gerdes & Zeh, 2009). Laser-induced elemental fractionation and instrumental mass discrimination were corrected by normalization to the reference zircon GJ-1 (0.0982 ± 0.0003; ID-TIMS GUF value). Repeated analyses of the reference zircon Plesovice and BB-16 and reference monazite Itabe and Manangotry (Gerdes & Zeh, 2006) during the same analytical sessions yielded an accuracy of better 1%. All uncertainties are reported at the 2 SD level. CL imaging of the minerals could not be performed due to technical issues but backscattered electron (EBS) imaging was done instead (Fig. S6).

Palynology

A total of eleven samples were chosen for palynological analyses, one from trench 3, four from trench 3/3, and six from trench 4. First, 25 g of sediment were separated from each sample, washed under water and crushed into gravel-sized fragments (about 2 mm), immersed in 15% HCl, followed by 30% HF, and finally in warm 10% HCl. A 12 µm filter was used to isolate the palynomorphs from the coarser grains. Palynological preparations for palynofacies were directly mounted on slides, although samples for organic preparations were further exposed in HNO3 for two minutes. Observation of palynological preparations was carried out using a Zeiss optical microscope and microphotographs were taken with an Infinity X (Lumenera) camera using Deltapix software.

Results

U-Pb geochronology

206Pb/238U data obtained for the granite and two volcaniclastic samples are given in Table S2. 206Pb/238U ages of individual zircons and monazites range between 171.1 ± 1.5 Ma and 189.3 ± 1.5 Ma, with the highest age probability around 172–173 Ma (Fig. 3). Two age populations are recognized in the volcaniclastic deposits: an ‘old’ one (1) 183.8 ± 1.8–189.3 ± 1.5 Ma, and a younger one (2) 171.1 ± 1.5–177.6 ± 1.7 Ma. The granite has one age population, at 172.8 ± 1.6 Ma that is interpreted as its crystallization age. Results of 206Pb/238U dating of the three samples have been plotted on three distinct concordia curves (Figs. 4A–4C), one for each sample. Analyses of single zircons from T4-3, T3-7, and the granite, yield mean 206Pb/238U ages of: (1) 173.0 ± 1.6 Ma (mean square of weighted deviates, MSWD = 0.89), (2) 172.8 ± 1.5 Ma (MSWD = 1.1), and (3) 172.5 ± 1.6 Ma (MSWD = 0.95), respectively.

Palynology and paleobotany

The distribution of palynological taxa recovered from the three trenches examined is shown in Fig. 5. Two samples collected from trench 3/3 and two samples from trench 3 (Figs. 5G, 5I, 5K, and 5L) do not contain palynomorphs. This probably represents a taphonomic bias partly related to the coarse-grained character of the deposits, less favorable to the accumulation of thin-walled spores and pollen grains (Batten, 1996; Traverse, 1988). The eight remaining samples contain spore-pollen spectra mostly represented by poorly preserved gymnosperm pollen. The spore-pollen spectrum from trench 4 is more diverse, and all collected samples contained palynomorphs. Figure 6 shows selected palynomorphs from Kulinda deposits and all the taxa are listed with authors and year of publication in Table S3.

Bisaccate morphotypes include high percentages of Pseudopicea spp. (5–19%), Pseudopicea variabiliformis (8–12%), and Pseudopicea grandis (3–7%). Pollen morphotypes closer to recent forms are rare and represented by the genera Piceapollenites (1–1.5%) and Pinuspollenites (1–1.5%). Alisporites spp. and Podocarpidites spp. are also rare. Common components of the assemblage are Ginkgocycadophytus spp. (1–7 %) and Cycadopites spp. (0.5–2.5%). Classopollis pollen are rather scarce at Kulinda (1–5%). Spore abundance is variable but is in general lower than that for pollen. Spores are dominated by Stereisporites spp. (1–19%), Cyathidites australis (2.5–9%), and Cyathidites minor (1.5–4%).

Besides the major taxa listed above, the assemblage also includes rare occurrences of the pollen Alisporites bisaccus, Dipterella oblatinoides, Piceites podocarpoides, Protoconiferus funarius, Protopicea cerina, Protopinus subluteus, Pseudopiceae monstruosa, Pseudopinus spp, Dacrydiumites spp., Pinus divulgata, P. incrassata, P. pernobilis, P. subconcinua, P. vulgaris, Podocarpidites multesimus, P. major, Sciadopityspollenites multiverrucosus, and taxa of uncertain affinity such as Callialasporites dampieri and Podozamites spp. Rare specimens of spores include the lycopods Annulispora folliculosa, Densoisporites velatus, Leptolepidites verrucatus, Neoraistrickia aff. taylorii, Perotrilites sp., Polycingulatisporites triangularis, Retitriletes subrotundus, Undulatisporites pflugii, U. fossulatus, Uvaesporites scythicus, the ferns Dictyophyllidites equiexinus, Eboracia granulosa, Gleicheniidites sp., Leiotriletes nigrans, L. pallescens, L. selectiformis, L. subtilis, Osmunda papillata, Osmundacidites jurassicus, Salvinia sp., the bryophytes Stereisporites infragranulatus, the horsetail Equisetites variabilis, and Acanthotriletes spp. and Punctatosporites scabratus, both of unclear affinities. The assemblage also contains very rare trilete, zonate spores of rather simple morphology resembling the genera Couperisporites, Kraeuselisporites, or the taxon Aequitriradites norrisii (see Fig. 6U).

Figure 3 Probability curve based on the LA-ICP-MS data performed on zircons and monazites.

Two age populations (i.e., peaks) can be discriminated from this curve.

Figure 4 Concordia diagrams for the three samples collected at Kulinda.

(A) Zircons and monazites collected from the granite. (B) Zircons collected from a sample situated above bone bed 3 in trench 3. (C) Zircons collected from a sample situated below bone bed 4 in trench 4.

Figure 5 Palynomorph distribution in the Kulinda deposits.

Legend: 1—coarse sandstone/breccia, 2—sandstone, 3—siltstone, 4—laminated mudstone, 5—non-correlated part of the stratigraphic section, 6—bones, 7—iron-oxide nodules, 8—abundances of the palynomorphs.

Figure 6 Selected palynomorphs from the trench exhibiting the dinosaur bones and feather-like structures.

(A) Alisporites similis (Balme) Dettmann. (B) Piceapollenites mesophyticus (Bolchovitina) Petrosjanz. (C) Osmundacidites jurassicus (Kara-Mursa) Kuzitschkina. (D) Cyathidites minor Couper. (E) Protopinus subluteus Bolchovitina. (F) Pseudopicea variabiliformis Bolchovitina. (G) Biretisporites eneabbaensis Backhouse. (H) Podocarpidites rousei Pocock. (I) Ginkgocycadophytus sp. (J) Stereisporites bujargiensis (Bolchovitina) Schulz. (K) Pseudopicea grandis (Cookson) Bolchovitina. (L) Podocarpus tricocca (Maljavkina) Bolchovitina. (M) Leiosphaeridia sp. (N) Classopollis classoides Pflug. (O) Alisporites bisaccus Rouse. (P) Protoconiferus funarius (Naumova) Bolchovitina. (Q) Pseudopicea magnifica Bolchovitina. (R) Stereisporites granulatus Tralau. (S) Pinus divulgata Bolchovitina. (T) Leiotriletes subtilis Bolchovitina. (U) undetermined spore taxon resembling Aequitriradites norrisii Backhouse. (V) Stereisporites sp. (W) Tripartina variabilis Maljavkina. (X) Stereisporites incertus (Bolchovitina) Semenova. (Y) Leiotriletes sp. Scale bars = 20 µm.

Discussion

The Kulinda deposits were previously regarded as Late Jurassic to Early Cretaceous, based on palaeontological comparisons and on the relative position of the section within the Ukurey Formation (Kozlov, Zaikov & Karasev, 1998; Rudenko & Starchenko, 2010). The new U-Pb radiometric investigations obtained here from Kulinda deposits provide direct evidence that can constrain the age of the locality.

Given the sedimentary origin of the deposits, the radiometric analyses give an indication of their maximal age. The Concordia ages for zircons and monazites (Fig. 4) recovered from both the volcaniclastics and the granite suggest an Aalenian (Middle Jurassic) age. We interpret the older age population in the volcaniclastic sediments as possibly representing inherited zircons from the granite or detrital zircons coming from another granitic source not sampled here. Chemical analyses (more information available in the Supplemental Information) performed on rock samples collected from the stratigraphic section shows that the sedimentary deposits are very similar in composition (see Table S4 and Fig. S7), indicating a similar source for all the deposits. This is also evidenced by the rare earth element (REE) pattern between the deposits and the granitic basement (see Fig. S6). Even if no zircon older than 173.4 ± 3.7 Ma has been sampled in the present study in the granite, our data suggest that the deposits at Kulinda, including the bone beds, are composed of material reworked from the nearby granitoids, cropping out on top of the hill. When only looking at the youngest zircon population, the three samples display a concordant age within error at 172.5 ± 1.6; 173.0 ± 1.6 and 172.8 ± 1.5 Ma (Fig. 4). This also emphasizes the genetic relationship between the granite and the volcaniclastic sediments from the bone beds. Consequently, the average age of 172.8 ± 1.6 Ma indicates the maximum age of the volcaniclastic sediments, which corresponds to the Aalenian.

The volcaniclastic origin of the deposits at Kulinda, together with the chemical and age for the granite suggest that the Kulinda deposits accumulated from the reworking of, at least, that Aalenian igneous source at the site of deposition. We cannot exclude the possibility that another igneous source yielded the older population of zircon, but this source must be chemically similar to the granite in any case. Because of the volcaniclastic nature of the Kulinda deposits, palynological information provides an essential complement to the radiometric data for refining the age of the deposits. The palynomorph distribution reflects the distribution of the taxa in the environment at the moment of sediment deposition whereas the absolute age of the zircons and monazites gives the crystallization age of the granitic source of the deposits.

Most of the palynomorphs recovered at Kulinda are characterized by wide stratigraphic ranges through both Jurassic and Cretaceous deposits (e.g., Norris, 1965; Cornet, Traverse & McDonald, 1973; Filatoff, 1975; Higgs & Beese, 1986; Ilyina, 1986; Markevich, 1995; Li & Batten, 2004; Pestchevitskaya, 2007; Ribecai, 2007; Markevich & Bugdaeva, 2009; Ercegovac, 2010; Lebedeva & Pestchevitskaya, 2012; Kujau et al., 2013; Zhang et al., 2014; Slater et al., 2018; Shevchuk, Slater & Vajda, 2018). The spore and pollen taxa observed in the Kulinda deposits are reported in Middle Jurassic palynozones that are calibrated to ammonite biozones from marine sections (Ilyina, 1986) and palaeofaunas from continental strata (Starchenko, 2010). In northern Siberia, the appearance of Pinus divulgata is attested in the Bajocian but the taxon is common elsewhere, from the Triassic through the Cretaceous; Alisporites bisaccus and Podocarpidites rousei are both reported in Bajocian deposits from Siberia and western Canada (Ilyina, 1986), although A. bisaccus has a much wider stratigraphic range and P. rousei is typical for the Bathonian of western Siberia and the Kansk-Achinsk Basin in southwestern Siberia (Krasnoyarsk region; Smokotina, 2006). The stratigraphically important taxa recovered from Kulinda deposits are Podocarpidites rousei, Eboracia torosa, and Gleicheniidites spp. These species are typical for the Bathonian in southern and northern regions of western Siberia (Ilyina, 1985; Shurygin et al., 2000) and Kansk-Achinsk Basin (Smokotina, 2006). Palynozone 10 from these regions includes Cyathidites spp., Sciadopityspollenites macroverrucosus, Eboracia torosa, and Classopollis, and a local palynozone includes Eboracia torosa, Quadraeculina limbata, and Classopollis. The strong domination of Pseudopicea variabiliformis is a characteristic feature of Bathonian assemblages, although this pollen is also abundant in older strata. Note that in western Siberia, the stratigraphic position of the palynozones is controlled by ammonites and foraminifers, and is therefore considered robust. Small pollen grains resembling Podocarpidites rousei are reported in assemblages from the middle part of the Bajocian of northern Siberia (Ilyina, 1985), but it is a characteristic pollen for Bathonian deposits in southern Siberia. Eboracia torosa is a Bathonian species (Ilyina, 1985), although the possibility that the taxon appeared earlier in other regions cannot be excluded. Gleicheniidites is an important component of the Kulinda assemblage. Its lowermost occurrences are defined in Bathonian sediments dated by macro- and microfauna in the East-European Platform (Starchenko, 2010; Mitta et al., 2012). This taxon has also been reported in Bathonian siliciclastic sediments from northeastern Ukraine (Shevchuk, Slater & Vajda, 2018). Another key feature is the low abundance of Classopollis spp., which excludes a Callovian age, characterized by high percentages of this pollen in the Siberian palaeofloristic region (Ilyina, 1985; Smokotina, 2006; Starchenko, 2010). It is interesting to note the presence of trilete zonate spores resembling Aequitriradites norrisii, which is an important species for the Middle Jurassic of Australia. Its lowermost occurrences are there revealed in the Bathonian, where the eponymous zone is defined for the middle part of this stage (Sajjadi & Playford, 2002). Nevertheless, an accurate determination of this taxon is not possible in the Kulinda material due to poor preservation.

Combination of the data recovered from the three examined trenches at Kulinda suggests that the deposits indicate a Bathonian age.

It should be noted that the macrofloral assemblages from Kulinda are similar in the different horizons where they were collected. Their taxonomic composition is characteristic for the Middle Jurassic—Early Cretaceous time range in Siberia, but do not provide more precise information about the age of the deposits. We will therefore not discuss this further in the main manuscript but a complete description of the macroflora is available in the Supplementary Material.

Conclusions

The deposits from the Kulinda section belong to the lower part of the Ukurey Formation, which crops out in several isolated depressions in the central and southeast Transbaikal region (Rudenko & Starchenko, 2010; Starchenko, 2010). The age of the Ukurey Formation was previously regarded as Late Jurassic to Early Cretaceous, based on biostratigraphic comparisons and local correlations. Previous radiochronological studies of volcanic rocks from the formation (with no clear location) indicated a Late Jurassic age (Rudenko & Starchenko, 2010; Starchenko, 2010). Our new results, combining absolute dating and palynological observations, place for the first time age constraints on the dinosaur-bearing volcaniclastics in Kulinda. The absolute dating of igneous and volcaniclastic rocks collected at Kulinda, indicate a maximal Aalenian age (172.8 ± 1.6 Ma) for the deposits that have yielded the Kulindadromeus fossils. Palynological data support a Bathonian age for the deposits, corresponding to an age ranging between 168.3 and 166.1 Ma (Gradstein et al., 2012), hence giving a minimum age. The stratigraphic range of the Ukurey Formation is therefore wider than previously assumed, its lower part extending to the Middle Jurassic. However, this new observation does not contradict the general geological framework of the region, characterized by marine deposits until the early Middle Jurassic, followed by continental sedimentation in grabens (Mushnikov, Anashkina & Oleksiv, 1966; Rudenko & Starchenko, 2010; Starchenko, 2010).

A Middle Jurassic age for Kulindadromeus is consistent with its phylogenetic position (see Fig. S8). A consistent and pectinate scheme of Middle Jurassic Asian basal neornithischians, including Agilisaurus louderbacki, Hexinlusaurus multidens and Kulindadromeus zabaikalicus, form a stem lineage culminating in Cerapoda (Godefroit et al., 2014): Parasaurolophus walkeri Parks, 1922, Triceratops horridus Marsh, 1889, their most recent common ancestor and all descendants (Butler, Upchurch & Norman, 2008). Agilisaurus and Hexinlusaurus were discovered in the lower member of the Shaximiao Formation of Dashanpu, Sichuan Province, China (Barrett, Butler & Knoll, 2005), and should therefore be Bajocian-Bathonian in age (Li, Yang & Hu, 2011). Yandusaursaurus hongheensis, from the upper member of the Shaximiao Formation of Dashampu (Bathonian-Callovian; Li, Yang & Hu, 2011) is not included in this analysis, but is also regarded as closely related to Agilisaurus and Hexinlusaurus (Boyd, 2015). In this phylogenetic scheme, Kulindadromeus is regarded as the sister-taxon of the vast clade Cerapoda. Cerapodan dinosaurs were particularly successful during the Cretaceous, being for example represented by pachycephalosaurs, ceratopsians, and iguanodontians (including the ‘duck-billed’ hadrosaurs). The earliest records of cerapodans are the dryosaurid iguanodontian Callovosaurus leedsi, from the Callovian of England (Ruiz-Omeñaca, Pereda Suberbiola & Galton, 2006), and the basal ceratopsian Yinlong downsi, from the Oxfordian of the Junggar Basin, Xinjiang, China (Xu et al., 2006). The calibrated phylogeny of ornithischian dinosaurs therefore suggests that cerapodans originated in Asia during the Middle Jurassic, from a common ancestor that closely looked like Kulindadromeus, then rapidly migrated to Europe, North America and Africa at the end of the Middle Jurassic and during the Late Jurassic.

Kulindadromeus is therefore the oldest known dinosaur with “feather-like” structures. The other Jurassic formations that have also yielded fossils of ‘feathered’ dinosaurs are younger. Recent U-Pb zircon CA-ID-TIMS data from Jianchang support a post-Middle Jurassic, Oxfordian (∼160 Ma), age for the Yanliao Biota preserved in the Lanqi/Tiaojishan Fm in western Liaoning (China; Li, Yang & Hu, 2011). Based on strong similarities of the fauna, together with available radioisotopic age evidence, it is generally accepted that the Lanqi Fm in Ningcheng (southeastern Inner Mongolia), and the Tiaojishan Fm in northern Heibei should be coeval with the Lanqi/Tiaojishan Fm in Jianchang (Zhou, Jin & Wang, 2010; Liu et al., 2012; Sullivan et al., 2014), thus also Oxfordian in age (Li, Yang & Hu, 2011). The lithographic limestones from Solnhofen and adjacent areas in South Germany that yield Archaeopteryx are early Tithonian in age (Schweigert, 2007).

The discovery of elongated and compound integumentary structures in the Middle Jurassic basal ornithischian Kulindadromeus will undoubtedly orient future research on the origin of feathers, which should be sought in much older deposits. If it can definitely be demonstrated that those structures are homologous to the feathers in theropods, the origin of feathers should be tracked back to the common ancestor of both dinosaur lineages (Godefroit et al., 2014) that most likely lived, regardless of the phylogenetic scenario considered for the relationships of the major dinosaur clades, during the Middle Triassic (Baron, Norman & Barrett, 2017).

Supplemental Information

Supplemental Information 1 Supplementary Information

Figure S5 photo credit: Aude Cincotta.

Click here for additional data file.

We would like to thank the reviewers for their helpful comments, which improved this paper. We gratefully thank Cyrille Prestianni and Paolo Spagna (RBINS) for the fruitful discussions and their help with various analyses. Gaëtan Rochez (University of Namur) and Thomas Goovaerts (RBINS) are thanked for their technical support. Maria McNamara (UCC) is warmly thanked for her advice and our fruitful discussions.

Additional Information and Declarations

Competing Interests

Author Contributions

Data Availability

The authors declare there are no competing interests.

Aude Cincotta conceived and designed the experiments, performed the experiments, analyzed the data, prepared figures and/or tables, authored or reviewed drafts of the paper, approved the final draft.

Ekaterina B. Pestchevitskaya performed the experiments, analyzed the data, prepared figures and/or tables, authored or reviewed drafts of the paper, approved the final draft.

Sofia M. Sinitsa approved the final draft, geology.

Valentina S. Markevich approved the final draft, palynology.

Vinciane Debaille contributed reagents/materials/analysis tools, reviewed the drafts of the paper, approved the final draft.

Svetlana A. Reshetova contributed reagents/materials/analysis tools, approved the final draft.

Irina M. Mashchuk and Andrei O. Frolov performed the experiments, approved the final draft, macroflora.

Axel Gerdes performed the experiments, contributed reagents/materials/analysis tools, prepared figures and/or tables, approved the final draft.

Johan Yans conceived and designed the experiments, approved the final draft and reviewed the drafts of the paper.

Pascal Godefroit conceived and designed the experiments, prepared figures and/or tables, authored or reviewed drafts of the paper, approved the final draft.

The following information was supplied regarding data availability:

The raw data is included in the Supplementary Tables.

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
