# Peer review of "The rise of feathered dinosaurs: Kulindadromeus zabaikalicus, the oldest dinosaur with ‘feather-like’ structures"

_PeerJ, doi:10.7717/peerj.6239_

## Round 0.1 · original submission · Minor Revisions

This manuscript provides important new age constraint data on an important fossil locality--this is a great addition to the scientific record! Based on feedback from the reviewers, required revisions are fairly minimal. Most critically, note requests from Reviewer 1 to provide a more detailed methodological explanation for the radiometric dating (e.g., grain selection, etc.). Also give consideration to any material in the supplemental text that could be moved into the manuscript. Finally, are the pollen samples archived anywhere, to permit future workers to verify observations and identifications here?

·

Basic reporting

This is a good manuscript with only a few issues that I can see. The writing is very clear and mostly free of errors or gramatical issues. The literature review is good and I cannot find any issues on this front.

The figures are not as good as one would hope. Figure 1B is very small and hard to see. It only shows what appears to be talus, not any stratigraphy. I would suggest combining figure 1A and Figure 2. I would remove Figure 1b and include it with Figure 3 instead. I would also like some high quality photos of each of the trenche sections. The stratigraphic column in Figure 3 is ok, but insufficient. I say this because there are inconsistencies in the manuscript over what was actually dated, which the methods stating volcaniclastics vs volcanic ash in quite ambiguous ways. I want to see photos of the ash beds and volcaniclastic deposits. This needs to be much more clearly stated. See comments in manuscrip and figure captions.

My number one concern is the apparently relaxed nature of the dating process. The authors state that monazite and zircon with dated, apparently together? and used to produce the reported ages. There is no mention of SEM or CL imaging used in the laser spot selection. This part of the paper needs to be cleaned up and clarified. If you are procuding ages from mixed populations of zircon and monazite, I want to know why and what different standards were used. Why are zircon ages and monazite ages not shown separately? This just needs a bit of cleaning up. I would like to understand what kind of screeing was used to select grains for dating, etc...

I found the conclusions well reasoned and solid.

In sum, good paper. Just needs a bit of tightening up.

Experimental design

Very good aims and scope.

The research questions and motivations are clear and valuable.

See points above regarding the methods. These are not as sound as they could be. Work is needed.

Validity of the findings

The only thing that I am not comfortable with is the dating approach combining monazite and zircon ages to apparently arrive at the reported concordia ages. This needs to be explained.

Otherwise the dating results look fine.

Additional comments

See comments above and in the attached manuscript.

·

Basic reporting

I have read and assessed the manuscript "The rise of feathered dinosaurs: Kulindadromeus
zabaikalicus, the oldest dinosaur with ‘feather-like’ structures." I have read the entire manuscript but will mainly restrict the feed-back on the palynology, which is my area of expertise.

This is an extremely well written manuscript, both concerning language and the organization of the paper. The manuscript is easy to follow, all through even for a non-vertebrate specialist. The palynological part is thorough, and the figures nice and necessary. It is difficult to date Jurassic terrestrial deposits by palynology but the authors have carefully built up their case.

The only minor issues that I can find is that Quadraeculina is misspelt (line 218) 8th letter should be a c instead of a q. I also miss a palynological photographical plate showing the main taxa – it is not crucial to include as the stratigraphical fig no. 6 is sufficient, but it would be nice to have one in the supplementary part, if possible.

Also some more recent relevant references are lacking:

Shevchuk, O., Slater, S. M., & Vajda, V. 2018. Palynology of Jurassic (Bathonian) sediments from Donbas, northeast Ukraine. Palaeobiodiversity and Palaeoenvironments, 98(1). https://doi.org/10.1007/s12549-017-0310-3
Slater, S. M., Wellman, C. H., Romano, M. & Vajda, V. 2018. Dinosaur-plant interactions within a Middle Jurassic ecosystem palynology of the Burniston Bay dinosaur footprint locality, Yorkshire, UK. Palaeobiodiversity and Palaeoenvironments, 98(1). https://doi.org/10.1007/s12549-017-0309-9
In summary, I suggest a very minor revision based on the palynological contribution. The rest of the manuscript is well-written and most probably correct but as I can´t judge this with certainty I leave those parts to other referees. I hope to see this manuscript in print as it contributes some interesting results.
Sincerely

Experimental design

Fine

Validity of the findings

Interesting with broad relevance.

Additional comments

See above.

---

## Round 0.2 · accepted · Accept

Thank you for your close attention to the comments from the reviewers, and for your thorough revision.

#